# Reliability of SNIP test and optimal number of maneuvers in 6-11 years healthy children

**Ana Aline Marcelino[1,2], Guilherme Fregonezi[1,2,3], Layana Marques[1,2‡], Ana Lista-Paz[4‡], Rodrigo Torres-Castro[3,5‡], Vanessa Resqueti[1,2,3] ***

**1** PneumoCardioVascular Lab/HUOL, Hospital Universitário Onofre Lopes, Empresa Brasileira de Serviços Hospitalares (EBSERH), Universidade Federal do Rio Grande do Norte, Natal, Rio Grande do Norte, Brazil, **2** Departamento de Fisioterapia, Laboratório de Inovação Tecnológica em Reabilitação, Universidade Federal do Rio Grande do Norte, Natal, Rio Grande do Norte, Brazil, **3** International Physiotherapy Research Network (PhysioEvidence), **4** Facultad de Fisioterapia, Universidade da Coruña, A Coruña, España, **5** Departamento de Kinesiología, Facultad de Medicina, Universidad de Chile, Santiago, Chile

☺ These authors contributed equally to this work.
‡ These authors also contributed equally to this work.
* vanessaresqueti@hotmail.com

## Abstract

### Background and purpose

Sniff nasal inspiratory pressure (SNIP) is a voluntary inspiratory maneuver measured through a plug occluding one nostril. The investigation of the number of maneuvers necessary to reach the highest peak of SNIP in pediatric populations has been inconsistent. Thus, this study aimed to assess the reliability of SNIP in healthy children aged 6 to 11 years according to sex and age group, and to determine the optimal number of SNIP maneuvers for this age group.

### Methods

This cross-sectional study included healthy children with normal pulmonary function. We performed 12 to 20 SNIP maneuvers, with a 30 s rest between each maneuver. The reliability was tested using the intraclass correlation coefficient (ICC), standard error of measurement (SEM), minimal detectable change (MDC), and Bland-Altman analysis for agreement.

### Results

A total of 121 healthy children (62 girls [51%]) were included in this study. The ICC and corresponding confidence interval (CI) between the highest measure and the first reproducible maneuver were 0.752 (0.656–0.824), SEM = 10.37 cmH$_2$O, and MDC = 28.74 cmH$_2$O. For children aged 6 to 7 years, the ICC was 0.669 (0.427–0.822), SEM = 10.76 cmH$_2$O and MDC = 29.82 cmH$_2$O; for children aged 8 to 11 years, the ICC was 0.774 (0.662–0.852), SEM = 9.74 cmH$_2$O, and MDC = 26.05 cmH$_2$O. For girls, the ICC was 0.817 (0.706–0.889), SEM = 9.40 cmH$_2$O and MDC = 26.05 cmH$_2$O; for boys, the ICC was 0.671 (0.487–0.798), SEM = 11.51 cmH$_2$O, and MDC = 31.90 cmH$_2$O. Approximately 80% of the total sample reached the highest SNIP before the 10[th] maneuver.

**Data Availability Statement:** All 'Reliability of SNIP test in healthy children' files are available from the Kaggle database (10.34740/kaggle/dsv/1539628).

**Funding:** This work was supported by: 1) Coordenação de Aperfeiçoamento de Pessoal de Nível Superior - Brazil (CAPES) - Financing Code 001; 2) GAFF, grant from CNPq number 312876/2018-1, Guilherme Augusto de Freitas Fregonezi; 3) VR, grant from CNPq number 315580/2018-6, Vanessa Resqueti. GAFF and VR had hole in study design, data analysis and revising the manuscript.

**Competing interests:** The authors have declared that no competing interests exist.

## Conclusions

SNIP demonstrated moderate reliability between the maneuvers in children aged 6 to 11 years; older children and girls reached the SNIP peak faster. Finally, results indicated that 12 maneuvers were sufficient for healthy children aged 6 to 11 years to achieve the highest SNIP peak.

## Introduction

Maximal static inspiratory and expiratory pressures (PImax and PEmax, respectively), and sniff nasal inspiratory pressure (SNIP) are noninvasive volitional tests commonly used to assess respiratory muscle strength in both adult and pediatric populations [1, 2]. These tests are highly dependent on individual effort and cooperation from the patient, your results vary widely, and low values may not necessarily reflect respiratory muscle weakness but a lack of motivation and poor coordination [3, 4]. Therefore, precise methods should be adopted during the assessment [1, 5]. Previous studies have determined the optimal number of maneuvers for PImax [6] and SNIP [3].

Accurate respiratory parameters may be used as endpoints for decision making in clinical practice [7]. Studies have shown that SNIP is a promising surrogate measure for the assessment of respiratory function in childhood neuromuscular disorders, such as Duchenne muscular dystrophy [7–9] and in adult neuromuscular disorders, such as amyotrophic lateral sclerosis, which appears to be more feasible than PImax in those with advanced disease [10].

SNIP involves a short sharp, voluntary inspiratory maneuver measured through a plug occluding one nostril while the sniff is performed using the contralateral nostril [1, 11]. As a significant learning effect is associated with the performance of SNIP, several maneuvers are required, with a plateau usually reached after 5 to 10 measurements in adults [3]. Lofaso et al. [3] in a study involving healthy children and adults with a variety of neuromuscular and respiratory disorders suggested using more than 10 SNIP maneuvers when the values were slightly below normal or when SNIP was used to monitor a decline in the inspiratory muscle strength. However, the results of study investigating the number of maneuvers necessary to reach the highest peak of SNIP in pediatric populations have been inconsistent [3].

Hence, the aim of our study was to evaluate the reliability of SNIP in healthy children aged 6 to 11 years according to sex and age group, and to establish an optimal number of SNIP maneuvers for this population.

## Methods

### Study design

This cross-sectional study was a secondary analysis of previously published data regarding reference values for SNIP in healthy children by Marcelino et al. [11]. Participants were initially evaluated for pulmonary function using spirometry and then the maximum static respiratory pressures tests (PImax and PEmax) and finally SNIP, our main variable. These evaluations mentioned were performed by the same trained evaluator, in the entire sample. The study was approved by the Hospital Universitário Onofre Lopes, Universidade Federal do Rio Grande do Norte Research Ethics Committee under number 2.051.325. The parents of the children included in the study provided written informed consent, and all procedures were performed according to the principles of the Helsinki Declaration.

## Participants

The sample consisted of healthy children, aged 6 to 11 years, stratified by age group 6 to 7 years and 8 to 11 years, and recruited from public schools in the city of Natal/RN, Brazil. Children included in the study met the following inclusion criteria: normal pulmonary function (FVC and $FEV_1 > 80\%$ of predicted and $FEV_1/FVC > 70\%$) [12]; no history of respiratory, cardiac, cerebrovascular, and neuromuscular diseases; no history of influenza or nasal congestion in the past week or identified at the time of evaluation; no regular use of medications for allergy and corticosteroids or central nervous system depressants; no septal deviation (reported by parents or observed by a marked discrepancy of SNIP values between the nostrils); and no history of previous thoracoabdominal surgery requiring incision of the thoracic or abdominal cavities [13–15]. Children who failed to perform the tests correctly or refused to participate were excluded from the study.

## Outcome measures

**Spirometry and maximal static inspiratory and expiratory pressures.** Spirometry was performed using a Koko spirometer (nSpire Health Inc, Longmont, CO, USA) with the children in the sitting position, following recommendations of the American Thoracic Society/European Respiratory Society and Brazilian Society of Pneumology and Tisiology [13, 16]. To ensure that all children achieved normal values, prediction equations for healthy Brazilian children described by Mallozi and published by Pereira [16] were used.

A digital manometer (NEPEB-LabCare/UFMG, Brazil) was used to measure respiratory muscle strength, with the children in the sitting position. PImax was measured starting close to the residual volume, while PEmax was measured close to the total lung capacity [15]. At least three measurements of each maximum respiratory pressure were recorded, with a 1-min rest between each maneuver. If the final measurement was the highest, more measurements were performed until a lower value was recorded. The highest values of each maximum mean pressure were used in data analysis and were compared with the reference values reported in the literature [17].

**Sniff nasal inspiratory pressure.** SNIP was assessed using a silicone nasal plug connected to the manometer by a polyethylene catheter. All children were asked to place the plug in the nostril, through which air could pass freely, and were asked to further perform a short, sharp inspiratory effort through the nostril, in the sitting position and with the lips closed [1]. This maneuver was performed near the functional residual capacity, and a passive relaxation immediately after reaching the peak pressure was requested to be performed [11, 18]. All children performed 12 maneuvers, with a 30 s rest between each maneuver [3, 5]. However, if the 11th or 12th maneuver was 10% higher than the highest of the first 10 measurements, recording was continued up to 20 measurements [11]. The following criteria were used to select SNIP values suitable and reproducible for the analysis: maneuver performed quickly and strongly; duration of SNIP $\leq 500$ ms; sustained pressure peak for $< 50$ ms; smooth and descending curve; and no biphasic peak [19]. The highest SNIP peak value was used for data analysis.

## Statistical analysis

The sample size was determined using the t-test based on the mean and SD of previous values found in the Stefanutti and Fitting [5] study. The sample size of 120 children was calculated based on the power of 99% and $\alpha = 0.05$ [11]. Data were analyzed using the GraphPad Prism software version 6.0 (GraphPad Inc., La Jolla, CA, USA). In the descriptive analysis, quantitative variables are expressed as mean ± SD or median and interquartile ranges, while categorical data are expressed as frequencies. The Kolmogorov-Smirnov test was used to assess the

normality of data distribution. The unpaired t-test or Mann-Whitney was applied for inter-group comparisons, as well as Spearman's $\rho$ for correlations between the highest SNIP peak position with the general characteristics of the sample. The Wilcoxon test was performed to compare the highest maneuver that a group of children reached after the 10[th] maneuver and the highest measurement achieved among the first 10. Differences with $p < 0.05$ were considered statistically significant. The intraclass correlation coefficient (ICC) was estimated using the Statistical Package for the Social Sciences software version 22 (IBM Corporation, Armonk, NY, USA) to analyze repeatability between the highest SNIP value and the first reproducible maneuver between sexes and age groups of 6–7 and 8–11 years. Values $< 0.5$ indicated poor reliability; between 0.5 and 0.75, moderate reliability; between 0.75 and 0.90, good reliability; and $> 0.90$, excellent reliability [20]. In addition, the standard error of measurement (SEM) was calculated between age groups and sexes as follows: SEM $= S \times \sqrt{(1 - ICC)}$, in which S corresponded to the highest SD obtained among the two tests compared [21]. The minimal detectable change (MDC) was calculated as follows: $MDC_{95} = SEM \times 1.96 \times \sqrt{2}$, in which 1.96 represents the level of confidence adopted (95%) and $\sqrt{2}$ corresponds to the correction factor for repeated measurements [21]. The Bland-Altman analysis was additionally performed between the two measurements specified above.

## Results

Initially, 139 children were recruited, but 121 healthy children (62 girls [51%] and 59 boys [49%]) included the study sample. Eighteen children were excluded for different reasons (deviated septum, spirometric changes, inability to perform the maneuvers, nasal congestion, and presence of asthma). Anthropometric, pulmonary function, and respiratory muscle strength data are summarized in Table 1. The ICC, SEM, and MDC between the highest maneuver achieved and the first acceptable measurement of the entire sample and that stratified according to sex and age group are reported in Table 2. The sample revealed moderate reliability between maneuvers and among the age group of 6 to 7 years and boys. Good reliability was

**Table 1. Anthropometric data and pulmonary function stratified by sex.**

|  | Total | Girls | Boys |
|---|---|---|---|
| **n (% total)** | 121 (100%) | 62 (51%) | 59 (49%) |
| **Weight (kg)** | 34.18 ± 10.18 | 33.40 ± 9.23 | 34.96 ± 11.12 |
| **Height (cm)** | 133.6 ± 9.68 | 133.5 ± 9.40 | 133.6 ± 10.05 |
| **BMI Percentile** | 70.87 ± 29.29 | 70.62 ± 27.20 | 71.12 ± 31.58 |
| **FVC (L)** | 1.95 ± 0.45 | 1.87 ± 0.41 | 2.04 ± 0.47 |
| **FVC (%)** | 105.1 ± 12.58 | 103.9 ± 13.08 | 106.4 ± 11.99 |
| **FEV$_1$ (L)** | 1.72 ± 0.34 | 1.67 ± 0.31 | 1.77 ± 0.36 |
| **FEV$_1$ (%)** | 100.1 ± 10.56 | 100.6 ± 11.62 | 99.65 ± 9.39 |
| **FEV$_1$/FVC$_{*100}$** | 88.73 ± 6.28 | 90.17 ± 5.70 | 87.21 ± 6.54 |
| **FEV$_1$/FVC (%)** | 95.41 ± 6.75 | 96.96 ± 6.13 | 93.77 ± 7.04 |
| **PImax (cmH2O)** | 90.31 ± 29.42 | 83.26 ± 27.17 | 97.73 ± 30.09 |
| **PImax (%)** | 102.9 ± 30.81 | 96.55 ± 28.73 | 109.5 ± 31.77 |
| **PEmax (cmH2O)** | 99.25 ± 26.21 | 94.73 ± 25.70 | 104.0 ± 26.11 |
| **PEmax (%)** | 101.9 ± 26.64 | 98.65 ± 25.40 | 105.4 ± 25.64 |
| **SNIP (cmH2O)** | 89.47 ± 20.25 | 91.11 ± 21.02 | 87.75 ± 19.43 |

Data presented in mean ± SD. BMI: body mass index; FVC: forced vital capacity; FEV$_1$: forced expiratory volume in one second; FVC $_{(\%)}$, FEV$_1$ $_{(\%)}$ and FEV$_1$/FVC $_{(\%)}$: percentage predicted by Mallozzi and published by Pereira [16]; PImax: maximal inspiratory pressure; PEmax: maximal expiratory pressure; PImax$_\%$ and PEmax$_\%$: percentage predicted by Lanza et al. [17]; SNIP: sniff nasal inspiratory pressure.

**Table 2. Reliability between the highest SNIP and the 1st acceptable stratified by sex and age groups.**

|  | ICC (CI) | SEM $_{cmH2O}$ | MDC $_{cmH2O}$ |
|---|---|---|---|
| **Total** | 0.752 (0.656–0.824) | 10.37 | 28.74 |
| **Girls** | 0.817 (0.706–0.889) | 9.40 | 26.05 |
| **Boys** | 0.671 (0.487–0.798) | 11.51 | 31.90 |
| **6–7 years** | 0.669 (0.427–0.822) | 10.76 | 29.82 |
| **8–11 years** | 0.774 (0.662–0.852) | 9.74 | 26.99 |

Reliability tested through variables ICC: intraclass correlation coefficient ($< 0.5$: poor reliability, 0.5–0.75: moderate reliability, 0.75–0.90: good reliability, $> 0.90$: excellent reliability) [20], SEM (standard error of measurement), and MDC (minimal detectable change). CI: confidence intervals.

observed between the age group of 8 to 11 years and girls. These results were affirmed from the analysis of the variables ICC, SEM and MDC.

The Bland-Altman concordance analysis of the total sample, stratified according to sex between these two maneuvers is shown in Fig 1. In this analysis, girls showed mean difference of 18.02 ± 12.86, with agreement limits equal to 43.22 and -7.18, while the boys showed a mean difference of 20.29 ± 16.13, with agreement limits of 51.90 and -11.32.

The percentage of children, according to the greatest maneuver achieved, is shown in Fig 2. A small majority of the sample reported the highest SNIP between the 6th and 10th maneuvers, while 80% of the total sample reached the highest SNIP before the 10th maneuver. Ten children (8.26%) needed to perform more than 12 SNIP maneuvers according to the methodological criteria, but only four (3%) reached the maximum peak after 12th. Children who reached the highest SNIP between the 11th and 12th maneuvers demonstrated a statistically significant difference ($p < 0.001$) compared with the highest SNIP reached before the 10th maneuver (Fig 3).

We performed comparisons of general characteristics among children who reached the highest SNIP peak on the first 10 maneuvers and after the 10th (Table 3). Furthermore, we also observed a weak negative correlation between the position number of the greatest maneuver with PImax ($\rho = -0.189$, $p = 0.04$) and percentage of predicted PImax ($\rho = -0.22$, $p = 0.01$).

## Discussion

Our study demonstrated that SNIP maneuvers studied in the total sample revealed moderate reliability. Stratified according to the age group, children aged 6 to 7 years also exhibited moderate reliability compared with good reliability in children aged 8 to 11 years. According to sex, boys exhibited moderate reliability and girls had good reliability. We observed that the repeatability to achieve the highest maneuver among children depends on sex and age group. Thus, we assumed that older children and girls start maneuvers with values closer to the SNIP peak attained, that is, a learning effect was observed quickly in these two cases. Older children probably due to their level of understanding reached the SNIP peak faster than younger children. We acknowledge that girls are likely to perform SNIP with greater ease, observed during the maneuvers. This probably led to greater ICCs in girls than those in boys and in older children, as confirmed by the lower SEM and MDC values and the Bland-Altman analysis. Previous studies have usually tested the reliability of SNIP at intervals of days or weeks. Barnes et al. [22] estimated the reliability of SNIP in healthy adults between sessions with a 1-week interval and obtained good reliability (ICC = 0.76, SEM = 11.94, and MDC = 33.10), while Nikoletou et al. [23] assessed SNIP repeatability in patients with chronic obstructive pulmonary disease in two sessions with a 3-week interval and reported an ICC of 0.94. Maillard et al. [24] evaluated the within-session reproducibility of SNIP in healthy adults and reported ICCs ranging

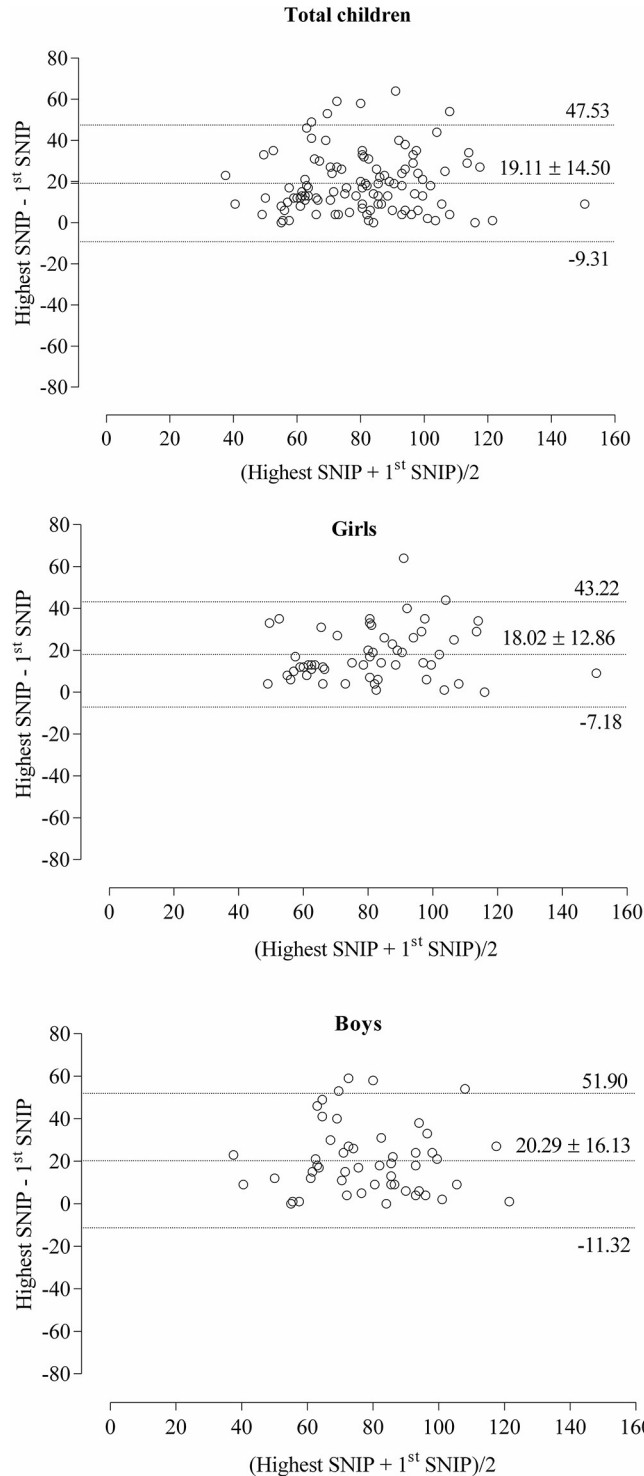

**Fig 1. Bland-Altman showing agreement between the highest SNIP (cmH$_2$O) and the 1$^{st}$ acceptable maneuver of total children and stratified by sex the middle line shows the mean difference.** The upper and lower limits of agreement represent 1.96 standard deviations above and below the mean difference.

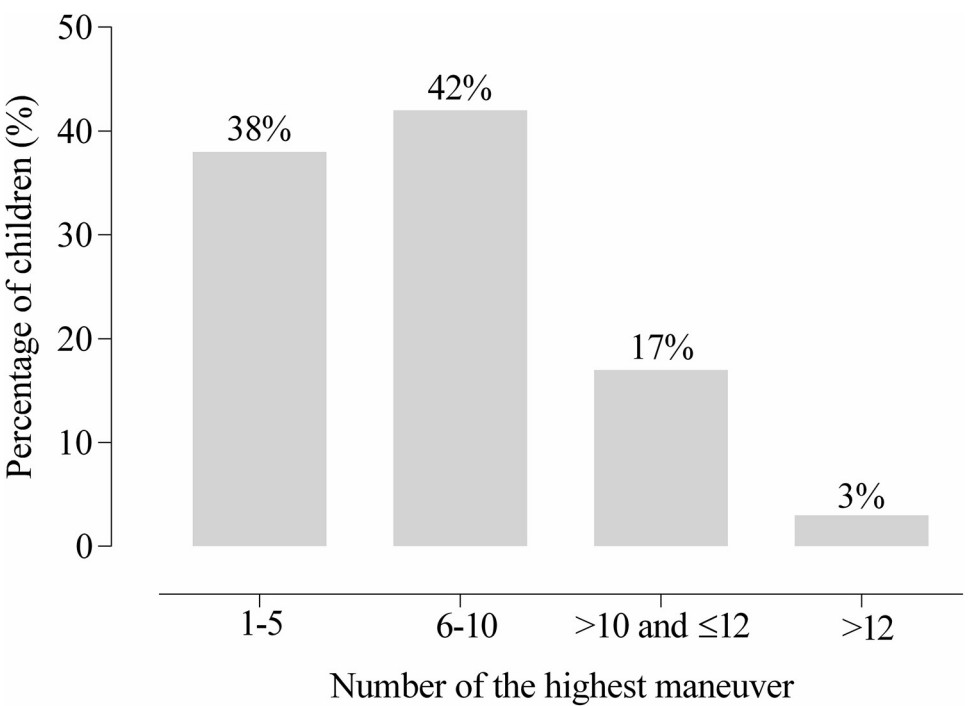

**Fig 2. Percentage of children by the number of highest maneuver.**

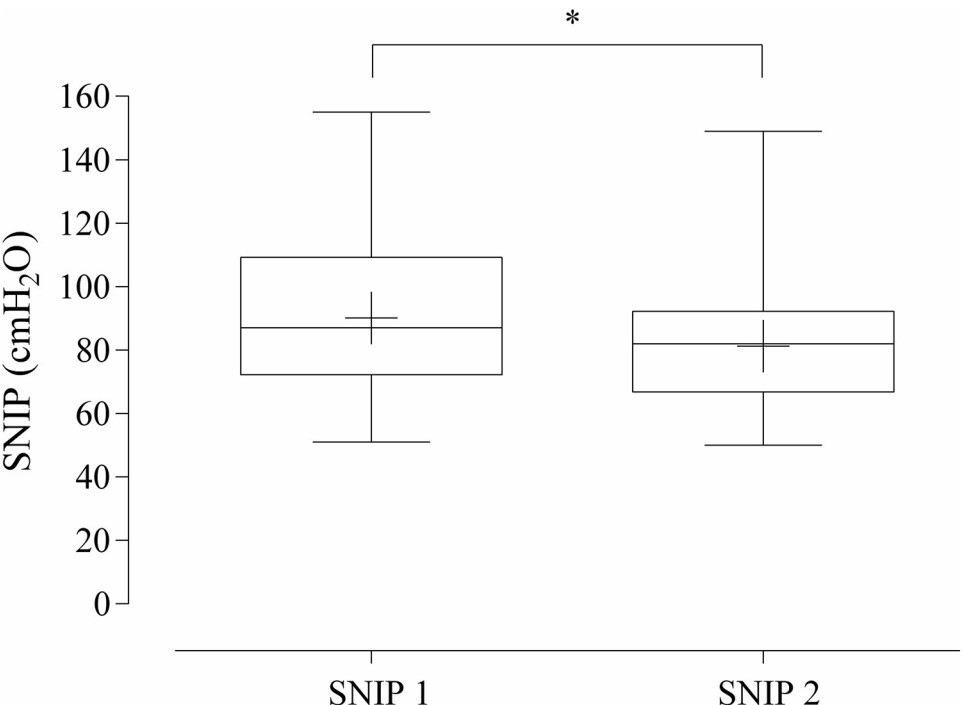

**Fig 3. SNIP maneuver (cmH₂O).** Box and Whiskers representing the comparison between the highest SNIP between the 11th and 12th maneuvers (SNIP 1) and the highest SNIP reached before the 10th maneuver (SNIP 2). + = SNIP mean; * p < 0.001.

**Table 3. Comparisons of general characteristics among children who achieved the highest SNIP until the 10th and after the 10th maneuver.**

| General features | ≤ 10 maneuvers | > 10 maneuvers | p |
|---|---|---|---|
| **n (%)** | 96 (79.3%) | 25 (20.7%) | - |
| **Age $_{(years)}$** | 8 (7–10) | 9 (7.5–10) | 0.68[a] |
| **Weight $_{(kg)}$** | 32.20 (26.78–38.65) | 36.10 (27.35–40.75) | 0.28[a] |
| **Height $_{(cm)}$** | 134.0 (125.0–140.0) | 137.0 (127.5–141.0) | 0.59[b] |
| **BMI percentile** | 78.65 (42.70–96.78) | 92.20 (58.25–98.95) | 0.19[a] |
| **FVC $_{(L)}$** | 1.94 (1.61–2.31) | 1.90 (1.59–2.35) | 0.90[b] |
| **FVC $_{(\%)}$** | 104.6 (96.81–113.1) | 105.5 (88.78–115.7) | 0.25[b] |
| **FEV$_1$ $_{(L)}$** | 1.73 (1.50–1.96) | 1.66 (1.40–2.08) | 0.99[b] |
| **FEV$_1$ $_{(\%)}$** | 100.5 (93.44–107.2) | 99.18 (90.27–105.7) | 0.19[b] |
| **FEV$_1$/FVC$_{*100}$** | 88.31 (84.85–92.85) | 90.12 (83.01–95.74) | 0.75[b] |
| **FEV$_1$/FVC $_{(\%)}$** | 94.96 (91.24–99.83) | 96.91 (89.26–102.9) | 0.75[b] |
| **PImax $_{(cmH2O)}$** | 91.50 (70.75–116.5) | 77.0 (59.50–98.50) | 0.017*[b] |
| **PImax $_{(\%)}$** | 102.9 (84.38–130.9) | 87.24 (72.92–104.0) | 0.012*[b] |
| **PEmax $_{(cmH2O)}$** | 95.0 (85.0–116.5) | 98.0 (80.0–126.5) | 0.84[a] |
| **PEmax $_{(\%)}$** | 100.6 (84.62–117.5) | 97.77 (83.06–124.9) | 0.78[a] |
| **SNIP $_{(cmH2O)}$** | 90.0 (74.50–100.8) | 86.0 (72.50–108.5) | 0.91[b] |
| **Highest SNIP peak position** | 6 (3–8) | 12 (11–12) | - |

Data presented in median and interquartile ranges (25–75%). BMI: body mass index; FVC: forced vital capacity; FEV$_1$: forced expiratory volume in one second; FVC $_{(\%)}$, FEV$_1$ $_{(\%)}$ and FEV$_1$/FVC $_{(\%)}$: percentage predicted by Mallozzi and published by Pereira (2002); PImax: maximal inspiratory pressure; PEmax: maximal expiratory pressure; PImax$_\%$ and PEmax$_\%$: percentage predicted by Lanza et al. (2015); SNIP: sniff nasal inspiratory pressure.

[a] Non-parametric test,

[b] Parametric test.

* $p < 0.05$.

between 0.85 and 0.92. These studies demonstrated good or excellent reliability in the populations studied, but both the population and the methodology were different, which limits additional comparisons with our results.

In the Bland-Altman analysis, we observed a slightly lower average difference in girls, which consequently, reduced the upper and lower concordance limits. Possible fatigue or discomfort is less likely to occur, because the maneuvers were performed according to the studies recommendations, with a rest interval of at least 30 s between each maneuver to prevent fatigue or discomfort. Barnes et al. [22] observed in their Bland-Altman analysis wide limits of agreement between days and between raters, which may be due to a number of limitations of the study including a potential learning effect that may have meant that with further repetitions the reliability may have increased.

In this study, we observed that most children regardless of age (42%), reached the highest SNIP between the 6th and 10th maneuver, as shown in Fig 2 and 80% reached the highest peak among the first 10 maneuvers performed. However, 17% of the total sample reached the highest SNIP between the 10th and 12th maneuvers, but only four children reached the maximum peak after 12 maneuvers. In addition to biological factors, such as age and sex, there was individual variation according to determinants, such as comprehension, learning time, and agility in performing the maneuver. As a result, some children achieved the highest maneuver within the first five attempts, and others needed more than 10 attempts. Hence, 12 maneuvers appear to be ideal for children in the described age group since most children do not improve the peak of the maneuver by performing more than 12 maneuvers. According to the American Thoracic Society [25], most individuals reach the SNIP plateau between five and 10

maneuvers. Uldry and Fitting [19] reported that a continuing learning effect was eliminated by the absence of further increments of SNIP during the final three maneuvers in the sitting position. Lofaso et al. [3] evaluated healthy volunteers and patients, including adults and children with various neuromuscular and pulmonary diseases, to investigate the highest SNIP improvement after the 10th maneuver. The authors found that a learning effect persisted after the SNIP, suggesting the need of additional sniffs when the best outcome of the first 10 maneuvers was slightly below normal. Terzi et al. [26] evaluated the learning effect and reproducibility of SNIP in healthy adults and observed that volunteers achieved the highest SNIP peak when performing an average of seven maneuvers, similar to the findings of our study. They determined that SNIP was less sensitive to the learning effect because higher SNIP values did not increase from session to session. PImax seems to influence the number of maneuvers to be developed. Children with higher PImax were able to reach the maximum peak of SNIP faster, which may be related to the level of muscle recruitment.

Lofaso et al. [3] observed that the majority of children with respiratory or neuromuscular disease were unable to perform a series of 20 SNIP maneuvers; hence, they suggested performing 10 maneuvers or more when possible. In healthy children, we observed that 20% of children reached the highest SNIP peak after the 10th maneuver, and only four reached the peak after the 12th maneuver. However, children that reached the highest SNIP between the 11th and 12th maneuvers demonstrated a statistically significant difference compared the highest maneuver reached before the 10th maneuver, suggesting that 12 and not 10 maneuvers were optimal In children with respiratory or neuromuscular diseases, these conditions can affect the optimal number of maneuvers and consequently, the reliability of the test. Nevertheless, studies have shown that SNIP is a well-tolerated maneuver that can be used to assess respiratory disorders [8, 27, 28]. In these specific cases, 12 maneuvers may also be sufficient, and when low values are reached, complementary assessments regarding PImax should be performed.

## Study limitations

Studies have generally demonstrated reproducibility between evaluations performed on different days and by different evaluators; however, we could not derive such inferences. The repeatability for this population between days and evaluators may be performed in future research. We believe that due to ethnic and racial variations, the results of our study should be interpreted and used with caution. Other objective data such as hemoglobin levels, which can affect SNIP, have not been measured but can be included in future research.

## Conclusions

The SNIP maneuver is a volitional test that measures inspiratory muscles strength. It is widely used to detect a decrease in the strength of these muscles, which may indicate muscle weakness or fatigue, when lower than expected. SNIP needs to be maximal and requires some practice to learn; hence, it is performed several times to achieve a plateau of pressure. Therefore, we conclude that the SNIP test demonstrated moderate and good reliability between the maneuvers, both general and stratified according to age and sex. We assume that older children and girls reach the SNIP peak quickly, that is, they learn to execute the maneuver in a shorter period, observed in the reliability analysis performed. Furthermore, we conclude that 12 maneuvers are sufficient for healthy children aged 6 to 11 years to achieve the highest SNIP peak.

## Author Contributions

**Conceptualization:** Guilherme Fregonezi, Vanessa Resqueti.

**Data curation:** Ana Aline Marcelino.

**Formal analysis:** Ana Aline Marcelino.

**Funding acquisition:** Guilherme Fregonezi, Vanessa Resqueti.

**Investigation:** Ana Aline Marcelino.

**Methodology:** Ana Aline Marcelino, Guilherme Fregonezi, Layana Marques, Ana Lista-Paz, Rodrigo Torres-Castro, Vanessa Resqueti.

**Supervision:** Ana Aline Marcelino, Guilherme Fregonezi, Vanessa Resqueti.

**Writing – original draft:** Ana Aline Marcelino.

**Writing – review & editing:** Guilherme Fregonezi, Layana Marques, Ana Lista-Paz, Rodrigo Torres-Castro, Vanessa Resqueti.

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
