## [Decision Letter · Decision Letter 0]

19 Mar 2021

PONE-D-20-31533

Reliability of SNIP test and optimal number of maneuvers in 6-11 years healthy children.

PLOS ONE

Dear Dr. Resqueti,

Thank you for submitting your manuscript to PLOS ONE. After careful consideration, we feel that it has merit but does not fully meet PLOS ONE’s publication criteria as it currently stands. Therefore, we invite you to submit a revised version of the manuscript that addresses the points raised during the review process.

We look forward to receiving your revised manuscript.

Kind regards,

Dragan Mirkov, Ph.D.

Academic Editor

PLOS ONE

Journal Requirements:

Reviewers' comments:

Reviewer's Responses to Questions

**Comments to the Author**

1. Is the manuscript technically sound, and do the data support the conclusions?

Reviewer #1: Yes

2. Has the statistical analysis been performed appropriately and rigorously? 

Reviewer #1: Yes

3. Have the authors made all data underlying the findings in their manuscript fully available?

Reviewer #1: Yes

4. Is the manuscript presented in an intelligible fashion and written in standard English?

Reviewer #1: Yes

5. Review Comments to the Author

Reviewer #1: Sniff nasal inspiratory pressure (SNIP) measurement is a volitional non invasive assessment of inspiratory muscle strength. In this cross sectional study the authors have studied 121 healthy children aged 6 to 11 years, to assess the reliability of SNIP according to age and sex and to determine the optimal number of SNIP manoeuvres for this age group. The study has a comprehensive inclusion and exclusion criteria. Standard recommendations to record PFT and SNIP were strictly adhered to.

Comments

1. SNIP manoeuvre is highly dependent on individual effort and cooperation. The instructor/evaluator of the test has significant influence on the maximum SNIP value attained and the number of attempts required to attain that maximum value. It is not mentioned whether the instructor/evaluator was same in all the subjects.

2. Having an objective assessment or score for level of motivation and co-ordination during the test would have added more value to the study.

3. We know that there would be significant ethnic and racial variation in PFT recordings. The same might be true for SNIP manoeuvre. Since this is a single center study, it results may not be generalisable.

4. Information regarding number of children excluded and the reason for exclusion are not clearly mentioned in the study.

5. It would be interesting to know if differences in anthropometry, pulmonary function and respiratory muscle strength among the included children would have altered (confounder) the number of attempts required to attain maximum SNIP. It is expected that children who required lesser number of attempts to attain maximum SNIP value would have had better anthropometry, pulmonary function and respiratory muscle strength.

6. It has been mentioned that studies recommend a gap interval of 30s between each manoeuvre. However 30s seems to be less for children and not clear if it would have led to early fatigue and sub-optimal peak SNIP value. It would be better if the authors provide a strong scientific basis for the same.

7. Further stratification of 8-11 years group into 8-10 years and 10-11 years could yield better intra-class correlation co-efficient, standard error of measurement and minimal detectable change in the latter group.

8. History and PFT alone might not exclude underlying diseases. Whether focussed clinical examination was done to look for any systemic illness needs to be mentioned.

9. Baseline characteristics including haemoglobin levels, serum electrolytes which could affect SNIP values should also have been measured in the sample population.

10. Further clinical assessment and investigation of those children who required more than 12 manoeuvres to attain maximum SNIP would have been interesting to know.

Kindly acknowledge the limitations of the study

6. PLOS authors have the option to publish the peer review history of their article (what does this mean?). If published, this will include your full peer review and any attached files.

Reviewer #1: No

---

## [Author Response · Author response to Decision Letter 0]

31 Mar 2021

Thank you for revise our manuscript. We appreciate the reviewers for their complimentary comments and suggestions. We have revised the manuscript accordingly to the recommendations. 

Please find attached a point-by-point response to reviewer’s concerns. We hope that you find our answers satisfactory, and that the manuscript is now acceptable for publication.

Sincerely,

Authors.

Reviewer #1:

1. SNIP manoeuvre is highly dependent on individual effort and cooperation. The instructor/evaluator of the test has significant influence on the maximum SNIP value attained and the number of attempts required to attain that maximum value. It is not mentioned whether the instructor/evaluator was same in all the subjects.

Answer: Thank you for your rectification. We add this information on study design section.

2. Having an objective assessment or score for level of motivation and co-ordination during the test would have added more value to the study.

Answer: Thanks for your comments. No objective assessment was developed for this purpose, but the same trained evaluator carried out all evaluations, as previously mentioned, ensuring the same motivation for everyone. 

3. We know that there would be significant ethnic and racial variation in PFT recordings. The same might be true for SNIP manoeuvre. Since this is a single center study, it results may not be generalizable.

Answer: Thank you for this consideration. Our study reveals this limitation, developed in one center in Brazil only. Nevertheless the sample size that is significant, we agree that ethnic and racial variation can be present. For this reason we include this information as a limitation of the study.

4. Information regarding number of children excluded and the reason for exclusion are not clearly mentioned in the study.

Answer: Thank you for the suggestion. We add this information on results section.

5. It would be interesting to know if differences in anthropometry, pulmonary function and respiratory muscle strength among the included children would have altered (confounder) the number of attempts required to attain maximum SNIP. It is expected that children who required lesser number of attempts to attain maximum SNIP value would have had better anthropometry, pulmonary function and respiratory muscle strength.

Answer: We appreciated this recommendation. According to your suggestion, we performed comparisons of the percentile, lung function (FVC and FEV1), and respiratory muscle strength (PImax, PEmax and SNIP) among children who reached the peak of SNIP between the first 10 and after the 10th maneuver, whereas in the literature we frequently observe the performance of 10 maneuvers on average. We add this information on Table 3 inserted on results section, and supplementary informations on statistical analysis, results e discussion sections. We observed that only PImax seems to influence the number of SNIP maneuvers. 

6. It has been mentioned that studies recommend a gap interval of 30s between each manoeuvre. However 30s seems to be less for children and not clear if it would have led to early fatigue and sub-optimal peak SNIP value. It would be better if the authors provide a strong scientific basis for the same. 

Answer: Thanks for the considerations. Some studies in adults and children (1 - Lofaso F, Nicot F, Lejaille M, Falaize L, Louis A, Clement A, et al. Sniff nasal inspiratory pressure: what is the optimal number of sniffs? Eur Respir J. 2006; 27:980-982. DOI: 10.1183/09031936.06.00121305; 2- Stefanutti D, Fitting JW. Sniff Nasal Inspiratory Pressure: Reference Values in Caucasian Children. Am J Respir Crit Care Med. 1999; 159(1):107-111. DOI: 10.1164/ajrccm.159.1.9804052) report SNIP with 30 seconds to rest between maneuvers. The literature already makes it clear that SNIP is a quick test, easy to perform, and does not cause muscle fatigue. Also, none of our assessed individuals reported any symptoms during data collection and the data show that the SNIP values, in most subjects, increase with each maneuver or maintaining similar values with slight variation (form more or less). We added two references to the manuscript to support this data on outcome measures section.

7. Further stratification of 8-11 years group into 8-10 years and 10-11 years could yield better intra-class correlation co-efficient, standard error of measurement and minimal detectable change in the latter group.

Answer: We appreciate the suggestion. We chose to present the results of the ICC only in two groups (6-7 and 8-11 years old) since we have less children in the 10-11 year group, therefore, if we consider three groups we can underestimate the values found in group 10-11 years and perhaps made some erroneous conclusion. Only to your appreciation, the ICC found when we separated were: 8-9 years: ICC: 0.830 (0.707 – 0.904), SEM: 10.32 and MDC: 28.60; 10-11 years: ICC: 0.654 (0.389 – 0.819), SEM: 11.45 and MDC: 31.73. 

8. History and PFT alone might not exclude underlying diseases. Whether focused clinical examination was done to look for any systemic illness needs to be mentioned.

Answer: Thank you for your observation. According to the inclusion criteria, we asked the children's parents about the presence or absence of diseases that could influence the SNIP results. Some children were excluded from the sample because they had asthma or nasal congestion during the evaluation and on the previous week. Besides, the pulmonary function test contributed to a greater reduction in the possibility of negative influence on maneuvers.

9. Baseline characteristics including hemoglobin levels, serum electrolytes which could affect SNIP values should also have been measured in the sample population.

Answer: Thank you for this valuable suggestion. Unfortunately, this assessment was not conducted. This proposal is extremely valid. In the future research we will consider included these baseline characteristics to observe the influence of these variables on SNIP measurement.

10. Further clinical assessment and investigation of those children who required more than 12 manoeuvres to attain maximum SNIP would have been interesting to know.

Answer: Thank you for your comment. According to the methodology considered, only ten children (8.26%) needed to perform more than 12 maneuvers and only four (3%) reached the maximum SNIP peak after the 12th maneuver. We considered a small number to observe significant clinical differences compared to the others in the sample. We recognize this results and this information was ratified on results section.

- Kindly acknowledge the limitations of the study.

Answer: As a suggestion, we recognized other limitations of our study and added them on study limitations section.

---

## [Decision Letter · Decision Letter 1]

11 May 2021

Reliability of SNIP test and optimal number of maneuvers in 6-11 years healthy children.

PONE-D-20-31533R1

Dear Dr. Resqueti,

We’re pleased to inform you that your manuscript has been judged scientifically suitable for publication and will be formally accepted for publication once it meets all outstanding technical requirements.

Kind regards,

Dragan Mirkov, Ph.D.

Academic Editor

PLOS ONE

Additional Editor Comments (optional):

Reviewers' comments:

Reviewer's Responses to Questions

**Comments to the Author**

1. If the authors have adequately addressed your comments raised in a previous round of review and you feel that this manuscript is now acceptable for publication, you may indicate that here to bypass the “Comments to the Author” section, enter your conflict of interest statement in the “Confidential to Editor” section, and submit your "Accept" recommendation.

Reviewer #1: All comments have been addressed

2. Is the manuscript technically sound, and do the data support the conclusions?

Reviewer #1: Yes

3. Has the statistical analysis been performed appropriately and rigorously? 

Reviewer #1: (No Response)

4. Have the authors made all data underlying the findings in their manuscript fully available?

Reviewer #1: (No Response)

5. Is the manuscript presented in an intelligible fashion and written in standard English?

Reviewer #1: Yes

6. Review Comments to the Author

Reviewer #1: (No Response)

7. PLOS authors have the option to publish the peer review history of their article (what does this mean?). If published, this will include your full peer review and any attached files.

Reviewer #1: No

---

## [Editor Report · Acceptance letter]

14 May 2021

PONE-D-20-31533R1 

Reliability of SNIP test and optimal number of maneuvers in 6-11 years healthy children. 

Dear Dr. Resqueti:

I'm pleased to inform you that your manuscript has been deemed suitable for publication in PLOS ONE. Congratulations! Your manuscript is now with our production department. 

Kind regards, 

on behalf of

Dr. Dragan Mirkov 

Academic Editor

PLOS ONE